# Multi-Omics Techniques in Genetic Studies and Breeding of Forest Plants

**Mingcheng Wang** [1,2,*], **Rui Li** [2,3] **and Qi Zhao** [2,3,*]

1   Institute for Advanced Study, Chengdu University, No. 2025 Chengluo Road, Chengdu 610106, China
2   Engineering Research Center of Sichuan-Tibet Traditional Medicinal Plant, Chengdu 610106, China; lirui@cdu.edu.cn
3   School of Food and Biological Engineering, Chengdu University, Chengdu 610106, China
*   Correspondence: wangmingcheng@cdu.edu.cn (M.W.); zhaoqi@cdu.edu.cn (Q.Z.)

**Abstract:** In recent years, the ecological and economic values of forest plants have been gradually recognized worldwide. However, the growing global demand for new forest plant varieties with higher wood production capacity and better stress tolerance cannot be satisfied by conventional phenotype-based breeding, marker-assisted selection, and genomic selection. In the recent past, diverse omics technologies, including genomics, transcriptomics, epigenomics, proteomics, and metabolomics, have been developed rapidly, providing powerful tools for the precision genetic breeding of forest plants. Genomics lays a solid foundation for understanding complex biological regulatory networks, while other omics technologies provide different perspectives at different levels. Multi-omics integration combines the different omics technologies, becoming a powerful tool for genome-wide functional element identification in forest plant breeding. This review summarizes the recent progress of omics technologies and their applications in the genetic studies on forest plants. It will provide forest plant breeders with an elementary knowledge of multi-omics techniques for future breeding programs.

**Keywords:** genetic breeding; high-throughput omics; multi-omics integration; gene regulatory networks; functional element identification

## 1. Introduction

Forest ecosystems, the major component of the terrestrial ecosystems in the biosphere, cover more than 30% of the global land area [1] and provide multiple ecosystem services with high ecological and economic values [2]. Forest plants are the key players in the forest ecosystem, providing suitable habitats for forest biodiversity while playing critical roles in the climate regulation, soil and water conservation, and hydrological and carbon cycles [3–5]. From an economic perspective, forest plants provide environmentally friendly products, including food, fiber, and fuel to fulfill human needs [6,7]. However, in the past few decades, increasing human activities and dramatic climate changes have caused remarkable biodiversity losses and reductions in forest cover, especially in tropical areas [8–10], weakening the functions and ecosystem services of global forests [11]. Currently, the native forests and forest plantations are threatened by multiple natural and anthropogenic stress factors, while the restoration of forest services require long periods [11]. Thankfully, the urgency to address forest health problems has been gradually recognized worldwide [10,11].

Since the mid-twentieth century, numerous genetic breeding programs have been conducted to reduce deforestation and forest degradation rates to develop new forest plant varieties with improved quality, productivity, and stress tolerance [12,13]. Conventional phenotype-based approaches, including hybrid, ploidy, and clonal breeding, have been successfully applied in several forest plants families, such as poplars [14], pines [15],

spruces [16], eucalypts [17], and larches [18], the major sources of wood worldwide. However, the genetic improvement of forest plants by conventional breeding is time-consuming, expensive, and inconvenient due to the long reproductive cycle, large body size, and complex reproductive process [13,19]. Usually, a forest tree breeding program using the conventional approaches includes several cycles of breeding and phenotypic selection (Figure 1A), which may take decades. Moreover, some important traits of forest plants, such as drought tolerance, growth, and branching, have low heritability; thus, their assessment in the field is difficult [13]. Therefore, conventional breeding approaches cannot simply satisfy the growing global demand for new forest plant varieties.

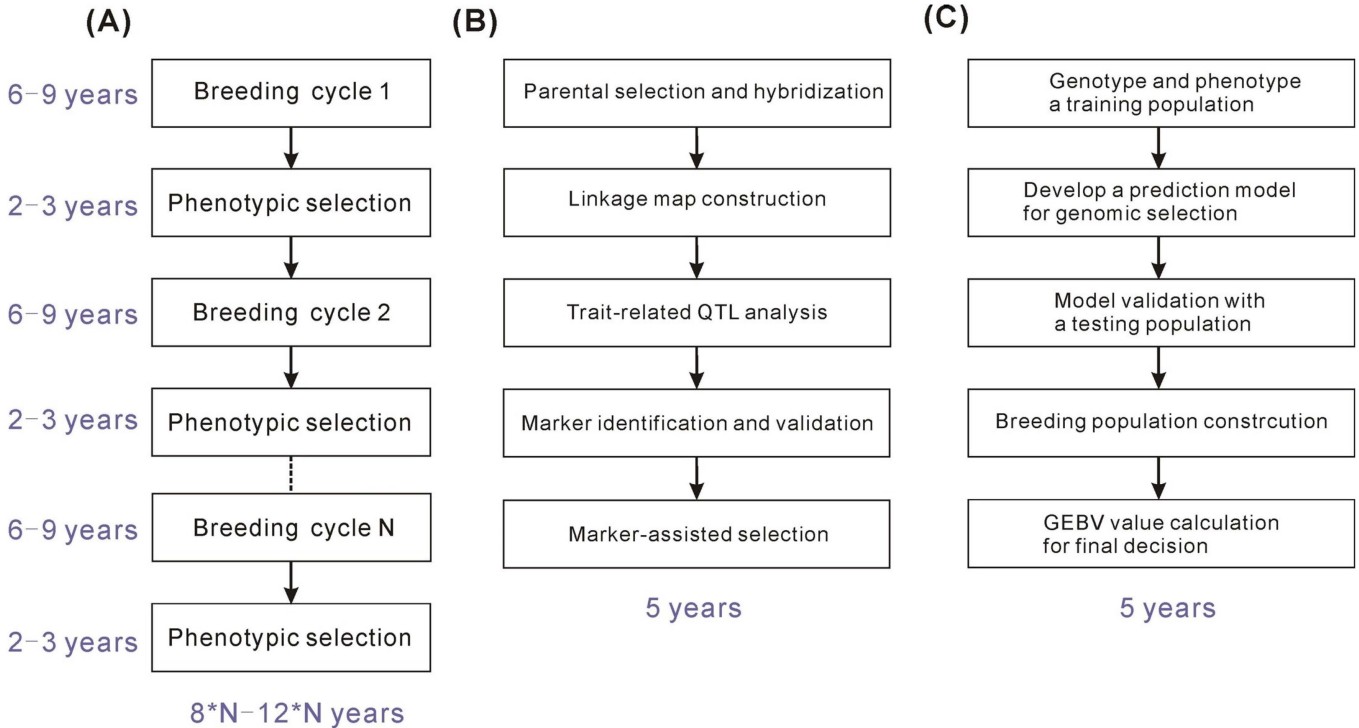

**Figure 1.** Basic procedures and time costing of breeding processes for *Eucalyptus* species: (**A**) conventional phenotype-based breeding; (**B**) marker-assisted selection; and (**C**) genomic selection.

With the rapid development of molecular genetics and sequencing technologies, scientists have invented "marker-assisted selection" (MAS), a more efficient and accurate approach, by integrating DNA markers into phenotypic selection [20,21]. MAS is an attractive tool for forest plant breeding as it allows individual selection based on genotypes instead of phenotypes (Figure 1B), which largely accelerates the breeding cycles by eliminating the long waiting period until plant maturity [13]. In MAS-based breeding, markers in strong linkage disequilibrium (LD) with trait-related quantitative trait loci (QTLs) are used as molecular tools [22]. However, although MAS has been successfully implemented in the genetic improvement of simple traits controlled by few QTLs with large effects, it has not effectively improved complex traits controlled by multiple QTLs with minor effects [23]. Genomic selection (GS), an improved form of MAS, uses genome-wide dense markers, such as single-nucleotide polymorphisms (SNPs), to maximize the minor effect of QTLs in LD with at least one marker [24]. During GS, the genomic estimated breeding value (GEBV) for each genotyped individual is calculated using a prediction model trained from the genotyped and phenotyped population by a joint marker analysis (Figure 1C), thereby avoiding potential bias in estimates and ensuring the overall accuracy of GS [25]. Compared to MAS, GS has significantly higher efficiency and accuracy for improving many traits in a wide range of species. However, the application of GS in forest plant breeding has been largely limited by the difficulty of genotyping extensive field plantations,

non-universality among populations, and high genotyping cost due to the lack of reference genome, particularly in conifers [13,23,26].

To date, several powerful tools for functional genomics, including clustered regularly interspaced short palindromic repeats (CRISPR)/Cas9 [27], RNA interference (RNAi) [28], and virus-induced gene silencing [29], have been extensively applied in the gene function studies of forest plant species. These advanced tools have provided opportunities to precisely engineer novel plant traits based on prior knowledge of their functional elements [30,31]. In addition, recent advances in high-throughput technologies using emerging omics approaches, including genomics, transcriptomics, epigenomics, proteomics, and metabolomics, have allowed a deeper dive into the molecular genetics of forest plant growth and development. These cutting-edge omics technologies provide powerful tools for understanding the underlying genetic mechanisms driving the complex architecture of various phenotypic traits, responses to biotic and abiotic stresses, and biosynthetic pathways of active compounds in forest plants. The increasing multi-omics data are also beneficial in the accurate genome-wide detection of functional elements using downstream bioinformatics analysis. Candidate functional elements identified through corroborating multi-omics evidence form a reliable basis for the genetic improvement of many growth traits and forest plant species. However, efficiently integrating large volumes of multi-omics data into forest plant genetic breeding programs is still challenging, especially for breeders with insufficient knowledge of omics and bioinformatics. This review discusses the recent progress of omics technologies, their applications in forest plant studies, and future application prospects of multi-omics approaches in the genetic improvement of forest plants.

## 2. Applications of Omics Technologies in Forest Plants

### 2.1. Genomics

The genome sequence of the model forest plant species, black cottonwood (*Populus trichocarpa*), was published in 2006 [32] and was recently updated to version 4.1 (Phytozome 13, release date: 5 October 2022) with significantly improved accuracy and continuity compared to the initial version. The release of the black cottonwood genome spawned research in the functional genomics of forest trees, especially angiosperms. However, conifers (gymnosperms), which dominate many temperate and boreal forests, have significantly different functional genomic characteristics from angiosperms, with unusually large genome sizes ranging from 18 to 35 gigabytes (Gb) [33]. In 2013, the first conifer genome belonging to Norway spruce (*Picea abies*) was published [34]. It was 4.3 Gb, only one-fifth of the Norway spruce actual genome size (19.6 Gb). However, the advent and development of next-generation sequencing (NGS) and third-generation sequencing (TGS) have greatly advanced the capacity to decode genomes of a wide range of forest plants. The emerging long-read and ultra-long-read sequencing and advanced assembly algorithms have provided useful tools for assembling complex plant genomes with high repeat content or heterozygosity (Figure 2) [35,36]. As a result, the difficulty of assembling a complete or near-complete genome of forest plants has been significantly alleviated in recent years [37]. To date, several high-quality forest plant genomes have been published, including gymnosperm species with genome sizes larger than 10 Gb, such as *Ginkgo biloba* [38], *Taxus chinensis* [39], *T. wallichiana* [40], *Larix kaempferi* [41], and *Pinus tabuliformis* [42].

A well-annotated reference-level genome provides a clear road map for downstream gene function and diversity studies, allowing an in-depth interpretation of genotype–phenotype relationships and functional DNA elements involved in various biological processes [43]. For example, the release of the black cottonwood genome has enabled studies on the genome-wide identification of regulatory genes and non-coding RNAs (ncRNAs) involved in several important biological processes, such as wood formation [44,45], annual growth cycle [46,47], flowering processes [48,49], and responses to abiotic stresses [50–52] in *Populus* species. Moreover, a comprehensive phylogenetic analysis of functional elements based on the genomes of several model plant species, including poplar, rice, and

*Arabidopsis* has been performed. A comparative genomic analysis among the different model species provides insights into the duplication history, selection pressure, and structural divergence of functional genes from an evolutionary aspect [53,54]. Therefore, the whole-genome-scale discovery and cross-species comparison of insertion and distribution of transposable elements, especially the long terminal repeat retrotransposons, could help further investigate their effects on the adjacent gene expression and plant phenotype [55,56]. Overall, the accumulation of high-quality genome sequences will provide a comprehensive understanding of the contribution of functional DNA elements to phenotypic variation among forest plants.

Many forest plant species have a wide geographical distribution, large population size, and high genetic diversity at local and regional scales [57]. Thus, a single reference genome cannot simply represent the DNA sequence diversity within a species. However, population genomics studies using whole genome resequencing (WGRS) or reduced-representation genome sequencing (RRGS) can provide insights into the genetic diversity of forest plants at the SNPs level. SNPs can be detected by mapping reads to a reference genome and subsequent variant calling based on WGRS at the population level. The detected SNPs allow further genome-wide polymorphism analysis during adaptive population divergence. For example, the WGRS of 427 moso bamboos (*Phyllostachys edulis*) from multiple representative geographic regions and subsequent population genomic analysis revealed several candidate genes under balancing selection or related to several agriculturally important traits, such as clear culm height, node number, density, and compressive strength [58]. In addition, several candidate genes related to light response, growth-promoting cytokinin, and wood development were identified by genome sequencing and WGRS of 80 silver birch (*Betula pendula*) with clear evidence of recent natural selection [59]. However, despite the continuously decreasing sequencing cost, the WGRS of many plant samples/species is still expensive. As an alternative or complementary approach to WGRS, RRGS consisting of reduced-representation libraries and restriction-site-associated DNA sequencing has been developed by integrating restriction enzymes into high-throughput sequencing to obtain a reduced genome representation [60]. Compared to WGRS, RRGS has apparent advantages of high efficiency, low cost, and does not require a reference genome [60]. RRGS enables genome-wide SNP discovery for non-model species lacking genome sequence information or species with large and complex genomes [61,62]. Furthermore, the genetic linkage map can be constructed using RRGS data. Subsequently, QTL mapping and genome-wide association analysis (GWAS) can be applied to identify the phenotype–genotype relationships across genomes of forest plants [63–65]. However, RRGS can also result in missing information possibly related to population differentiation, limiting its application scope.

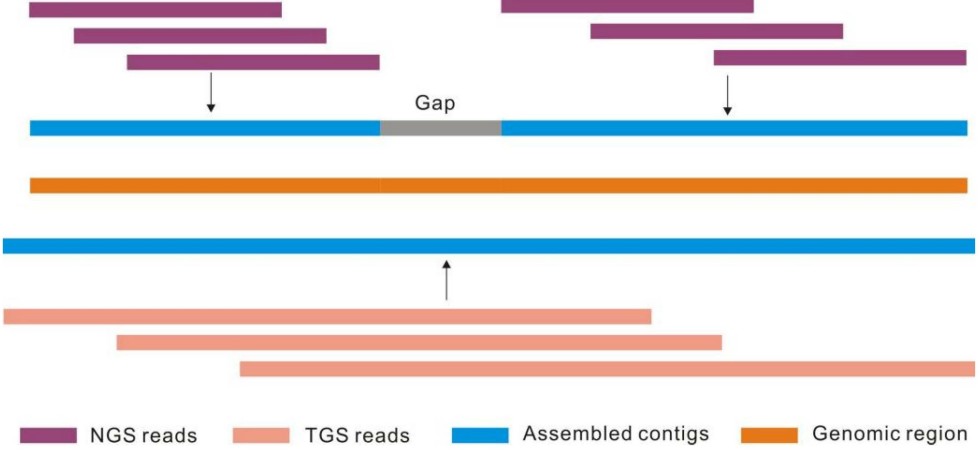

**Figure 2.** A schematic diagram showing the advantage of third-generation sequencing (TGS) reads in the assembly of highly complex genomic regions over next-generation sequencing (NGS) reads.

WGRS and RRGS easily detect SNPs and short insertions and deletions. However, structural variations (SVs), including the presence/absence variants (PAVs), copy number variants, and chromosomal rearrangements, are rarely detected by short-read sequencing [66]. SVs genetically control the phenotypic variability within and between plant species [67,68]. Recent advances in long-read sequencing have enabled the generation of high-quality assemblies for several individuals per species across many plant species, providing a solid foundation for accurately identifying SVs by pan-genomic analysis at the species level [66,69]. Therefore, a pan-genome represents a more comprehensive DNA sequence diversity of a plant species or taxonomic group. As a result, pan-genomic studies have been carried out to comprehensively understand the genetic diversity of several model plant species or economically important crops, including *Arabidopsis*, barley, rice, tomato, and soybean [70]. The release of these pan-genomes enabled more accurate associate mapping and population genomic analysis, identifying more potential genomic variants related to complex agronomic traits [65]. Population SVs identified in the pan-genomic studies also improved the GS-based prediction accuracy by partially rescuing the "missing heritability" of complex traits [65]. Furthermore, the pan-genomes provide a comprehensive list of candidate genes for direct genome editing, greatly accelerating the precision breeding of these crops. However, pan-genome research has been conducted on very few forest plant species, including poplar [71] and pecan [72]. Given the universality of sequencing technologies, assembly algorithms, and pan-genomic analysis pipelines, pan-genome will soon become a routine analysis tool for mining genetic variation and functional DNA elements of forest plant species.

In addition to genome sequences, population genomics, and pan-genomics, the three-dimensional genome structure, chloroplast genome, and mitochondrial genome are also useful for gene discovery and genetic engineering of plants [73–75]. With the increase in different genomic resources, integrating the existing genomic data will facilitate an efficient and accurate functional element identification of forest plants, which is beneficial in linking all DNA variations to phenotypes and designing molecular markers for further breeding processes. Genomic data from different sources have been submitted to publicly available databases, such as the National Center of Biotechnology Information, Ensembl, CoGe, GigaDB, and BIG Data Center. The development of a web-based, comprehensive genomic database, similar to BRAD [76] and Gramene [77], could also largely accelerate the genomic data integration for the genetic breeding of forest plant species. Currently, the applications of genomic technologies in forest plant breeding are limited by the complexity of polyploid genomes and the lack of genomic resources in some understudied species [69].

### 2.2. Transcriptomics

Transcriptomics is one of the most commonly used omics approaches in plant biology research. It involves studying the transcriptome, the complete set of transcripts generated by a cell or tissue [78]. Understanding the transcriptome is crucial in elucidating the structural and functional organization of the genome [79]. Several hybridization- and sequence-based approaches [79], including microarray, expressed sequence tag sequencing, serial analysis of gene expression, and RNA sequencing (RNA-seq), have been developed for transcriptome profiling. Among these approaches, RNA-seq, which captures all transcripts by high-throughput sequencing, is a revolutionary tool for accurate high-resolution transcriptome analysis [80]. For example, NGS-based RNA-seq generates millions of short reads, ranging between 25 and 300 bp in length.

Many computational tools have been developed to interpret the RNA-seq short-reads data. For example, a transcriptome assembly was reconstructed by aligning RNA-seq reads to a known genome assembly using assemblers such as Cufflinks [81], StringTie [82], and Scripture [83]. Furthermore, RNA-seq reads have been de novo assembled into transcripts even without a reference genome using assemblers such as Trinity [84], SOAPdenovo-Trans [85], and Oases-Velvet [86]. These reference-based or de novo strategies have been successfully applied in constructing a reference transcriptome with RNA-seq reads for many

plant species. However, although NGS short reads have high accuracy, they rarely span multiple exons. Furthermore, assembling NGS short reads into full-length transcripts is complicated by alternative splicing events frequently occurring in the genome [87]. Luckily, this challenge is alleviated by TGS-based transcriptome sequencing approaches, such as full-length isoform sequencing and nanopore-based direct RNA sequencing, which allows the direct sequencing of full-length transcripts without assembly [88,89]. However, given the relatively high error rate of TGS reads, highly accurate NGS RNA-seq reads are required to improve TGS-based transcriptome assembly accuracy [90]. At present, the high-quality and full-length transcriptome of forest plant species, including *Larix kaempferi* [91], *Chosenia arbutifolia* [92], *Fritillaria cirrhosa* [93], and *Alsophila spinulosa* [94] has been obtained.

The reference-level transcriptome assembly is crucial for the downstream analysis of gene expressions under multiple conditions (Figure 3). The gene expression levels are usually evaluated based on the RNA-seq reads. By mapping the RNA-seq reads to a reference transcriptome or genome, the number of reads matching each gene and gene expression levels are quantified by normalizing the read counts using algorithms, such as fragments per kilobase of mapped reads, transcripts per million and counts per million [95]. Subsequently, the differential expression (DE) analysis is performed using either non-parametric or parametric tools, such as DESeq2 [96], edgeR [97], and SAMseq [98]. DE analysis is widely used to analyze the forest plant responses to biotic and abiotic stresses, including drought, heat, salinity, flooding, cold, ultraviolet radiation, diseases, and insects [99,100]. For instance, the gene expression analysis in poplar root under polyethylene glycol-induced drought stress by TGS and NGS transcriptomic sequencing revealed several differentially expressed genes (DEGs) related to plant responses to drought in the biosynthesis and metabolism pathways [101]. In addition, the dual RNA-seq analysis of the interactions between Norway spruce and *Heterobasidion* revealed that several genes involved in the abscisic acid signaling were differentially expressed in Norway spruce, which might contribute to the Norway spruce response to the pathogen attack [102]. DE analysis also outlines the transcriptome dynamics across different tissues and plant developmental stages [103]. Notably, DE analysis under multiple conditions may identify too many or too few DEGs, requiring the analysts to integrate biological data from other sources or change the software parameters.

The gene expression data can also be used to construct the gene co-expression network (GCN), a powerful tool for the further elucidation of the gene regulatory relationships and identification of candidate functional genes [104]. The weighted gene co-expression network analysis (WGCNA) is a widely used pipeline to construct the GCN by clustering genes into modules based on their expression patterns and hub genes within each module [105]. The gene co-expression analysis has been successfully applied in detecting key functional genes regulating various biological processes in forest plant species, such as *Pinus tabuliformis* [106], *Populus trichocarpa* [107], *Hevea brasiliensis* [108], and *Zanthoxylum armatum* [109]. As a result, the cross-species GCN comparison effectively deduces the origin of new phenotypes and conserved gene functions at the species level [110]. Although this approach is rarely applied in forest plants, it has great potential in mining hub genes encoding important traits of forest plants. Moreover, transcriptome sequence and RNA-seq data are also applied in the genome-wide detection of SNP and simple sequence repeat markers (Figure 3), which are valuable molecular tools in forest plant breeding [111].

With the advances in RNA-seq and tissue processing approaches, single-cell RNA-seq (scRNA-seq) has become a revolutionary tool for studying plant functional genomics at the cellular level [112]. scRNA-seq can analyze any tissue in any plant. This emerging approach obtains the transcripts of thousands of cells per sample, providing new insights into gene expression heterogeneity across cells. Based on the scRNA-seq data, cells can be clustered into different categories using dimensionality reduction and clustering, which allows for the reconstruction of the cell differentiation trajectories [112]. Distinct expression patterns of different cell clusters provide further insights into the gene regulatory networks and candidate genes related to plant organ development. For instance, the scRNA-seq of 6796 poplar stem cells predicted the cell differentiation trajectories involved in phloem and

xylem development and candidate genes related to vascular development in poplars [113]. Another newly developed technique, spatial transcriptomics, quantifies and localizes gene expression within the tissue by combing histological imaging and RNA sequencing [114]. Its derivative, spatial single-cell transcriptomics, also recently developed, integrates the spatial information from spatial transcriptomics and cellular gene expression based on cRNA-seq, clearly elucidating the complex spatial gene regulatory networks related to plant organ development [115]. Despite its high costs, spatial single-cell transcriptomics has great potential in the precise breeding of forest plants.

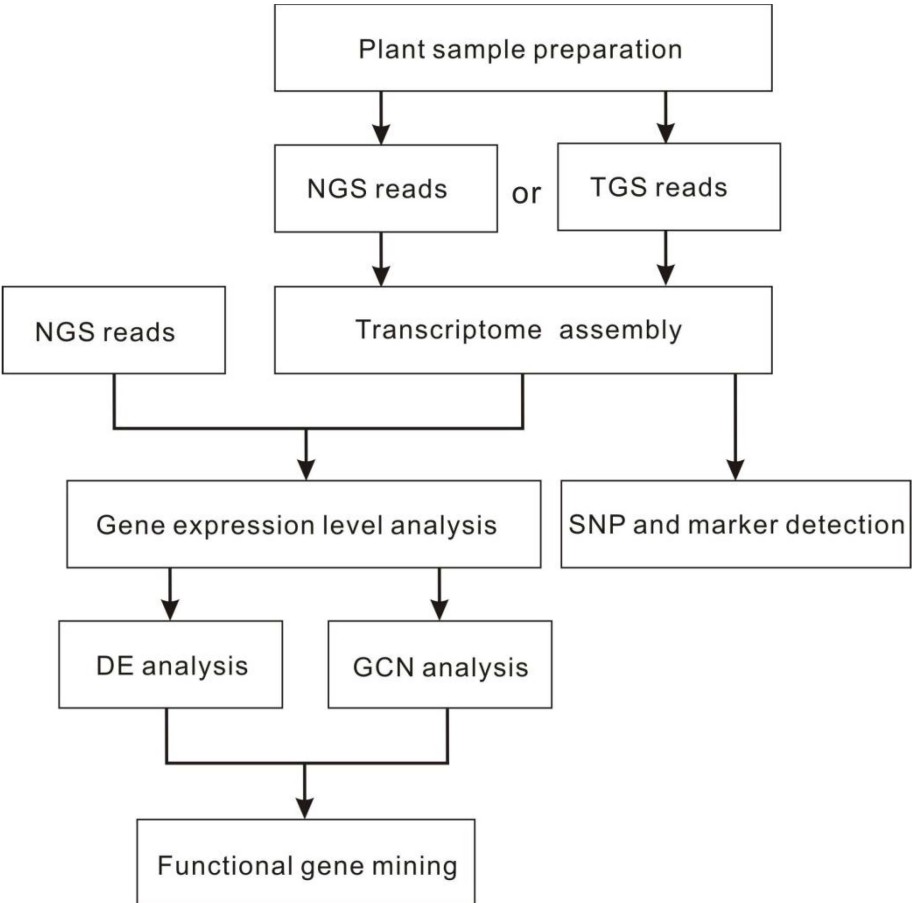

**Figure 3.** A commonly used workflow for the analysis of whole transcriptome sequencing data. DE, differential expression; GCN, gene co-expression network; SNP, single-nucleotide polymorphism.

Moreover, various transcriptomic technologies are beneficial for the deep mining of candidate resistance- and biosynthesis-related genes in forest plants. Subsequently, the functionally validated candidate genes can be used in developing new forest plant varieties with improved stress tolerance and increased bioactive compound contents. For example, a series of transcriptomic studies have revealed the important roles of several genes in poplar response to salt tolerance [116–118], which can be further applied in the transgenic breeding and phenotypic selection of salt-tolerant poplars. Although the price of transcriptome sequencing has significantly dropped, the need for fresh samples and complex stress response mechanisms have limited the application of transcriptomics in forest plants.

*2.3. Epigenomics*

Epigenomics studies the epigenome, which consists of the biochemical modifications in the nuclear DNA, histone proteins, and ncRNAs [119]. Although these epigenomic modifications do not alter the nucleotide sequences, they are inherited across generations

through mitosis or imprinting. Epigenetic changes, such as DNA methylation, histone modifications and variants, and ncRNA regulation, are frequently induced by environmental stresses or endogenous signals during plant development, which alters the chromatin structure and gene expression [120]. As a result, epigenetics is a powerful driving force in the environmental adaptation of plants by altering their phenotypic plasticity [119]. Therefore, epigenomic studies can provide important insights into the epigenetic basis underlying complex phenotypes and the local adaptation of forest plants, which cannot be deduced using DNA sequence variants.

The availability of high-quality reference genomes facilitates the genome-wide detection of epigenetic variants at the single-nucleotide level in forest plant species [121]. DNA methylation has been extensively studied in plants by detecting base modifications using bisulfite or long-read sequencing [122]. In plants, DNA methylation occurs as CG, CHG, and CHH in gene bodies and transposable elements. More importantly, the methylation levels substantially vary across plant species, tissues, and cells [123]. In addition, DNA methylation is a highly dynamic process during plant growth and development, including the establishment, maintenance, and active removal of methylation sites [123]. The dynamics of DNA methylation play important roles in the epigenetic regulation of plant growth, development, and response to environmental stresses [124]. For instance, several studies have revealed the potential role of DNA methylation in flower development [125], drought tolerance [126], wood formation [127], and immune response to pathogen infection [128] in *Populus* species. Moreover, the recently developed single-cell methylation profiling approaches have allowed the tracing of DNA methylation dynamics at the single-cell level [129].

In addition to the DNA methylation marks, histone marks, including histone modifications and variants, also transcriptionally silence or activate genes [130,131]. Histone modifications include histone methylation, acetylation, phosphorylation, ubiquitylation, sumoylation, and the reversible amino acid modifications at the N-terminal tail of histone proteins within the nucleosome core [120]. Among these modifications, histone methylation and acetylation are the most studied modifications regulating plant development and environmental stress response [132,133]. Histone variants are sequence variants of core histones H2A, H2B, H3, and H4, which regulate nucleosome structure and function [134]. Histone marks are detected by DNA/RNA-protein interactions across the genome using chromatin immunoprecipitation sequencing or the transposase-accessible chromatin assay with high-throughput sequencing [135].

Furthermore, regulatory ncRNAs, including the long non-coding and small RNAs, are important epigenetic marks with diverse functions in response to abiotic stress in forest plants [136]. For instance, several micro RNAs are differentially expressed during Norway spruce embryo development, potentially contributing to epigenetic memory and climatic adaptation [137]. Overall, the functional analysis of different epigenetic marks has extended the scope of plant biology.

Understanding the epigenetic mechanisms and variants is beneficial for the epigenetic improvement of forest plants. Naturally or artificially induced epigenetic variants serve as a novel genetic resource for plant epi-breeding [120]. Since the naturally occurring epigenetic variations are greatly limited, several laboratory-based approaches, including chemical treatment, biotic and abiotic stress treatment, tissue culture, grafting, RNAi, and direct epigenome editing by CRISPR/Cas9 have been applied to manually modify the plant epigenome [138]. These artificial methods are powerful tools in epi-breeding programs, which induce a wider range of phenotypic variation while increasing the transgenerationally inherited epialleles (Figure 4). Naturally and artificially induced epialleles can be employed as epigenetic markers in quantitative epigenetics based on the epigenetic quantitative trait loci and epigenome-wide association, which are important steps at the early stage of epi-breeding [139]. Moreover, epigenome editing methods using advanced CRISPR/Cas9 or CRISPR off technologies directly increase the stress resilience of plants through epigenome engineering [140,141]. In recent years, quantitative epigenetics and

epigenome engineering have been successfully applied in the epigenetic breeding of crop plants, including rice, tomato, potato, and soybean [142–145]. For example, Yu et al. (2021) modulated the plant RNA m$^6$A methylation using a transgenic approach, which significantly increased the rice and potato yield and biomass [144]. Compared to crops, the epigenetic mechanisms and variants in forest plants are relatively understudied. With an increased knowledge of the epigenetics of forest plants, epigenetic breeding will play an important role in improving more complex traits, promoting the forest plants' adaptation to the changing climate. However, given the complexity of the epigenetic mechanisms, the rapid development of new epigenetic markers, epigenome editing tools with improved efficiency, and deep epigenome sequencing of non-model forest plants are needed.

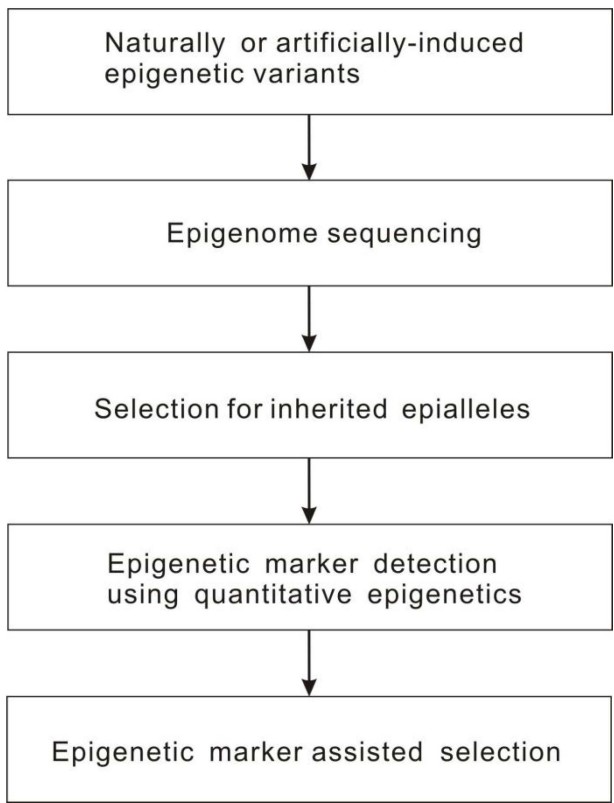

**Figure 4.** Basic procedure of epi-breeding using epigenetic markers.

*2.4. Proteomics*

Proteins are large biological molecules, the main undertakers of life activities, and are important components of plant cells and tissues, which form the physical basis of life. Proteomics studies the proteome, including the composition, localizations, modifications, and interactions of all the proteins expressed in a genome [146]. The plant proteome significantly varies across the cells and under different developmental and environmental conditions [147]. In addition, most eukaryotic proteins undergo post-translational modifications, altering protein expressions and functions [148]. Thus, proteomics is a powerful omics tool for comprehensively understanding the biological processes in the post-genomic era [149]. Genome sequencing provides DNA sequences of protein-coding genes, laying a solid foundation for proteomics research. The proteome contains much more complex functional gene information than that provided by the genome [150]. As a result, the accurate identification and quantification of the complete proteome are highly challenging. So far, several sequencing technologies have been developed for proteomics-based analysis (Figure 5), including protein microarrays, gel-based approaches, quantitative approaches (isobaric tags for absolute and relative quantification (iTRAQ), isotope-coded affinity tag, and stable isotope labeling with amino acids), and high-throughput approaches

(mass spectrometry and nuclear magnetic resonance spectroscopy) [146]. Among these approaches, liquid chromatography with tandem-mass spectrometry and matrix-assisted laser desorption/ionization-time-of-flight mass spectrometry is widely used to monitor plant proteome dynamics. A comprehensive plant proteome profiling provides valuable insights into the molecular mechanisms underlying plant growth, development, and stress responses [147]. For example, the tandem mass tag-based proteome sequencing of Masson pine (*Pinus massoniana*) with different resin yields revealed several differentially expressed proteins related to resinosis [151]. In addition, the *Picea asperata* somatic embryo proteome profiling using iTRAQ and comparative proteomics analysis under partial desiccation treatment provided novel insights into stress-related proteins and metabolic pathways in *P. asperata* [152].

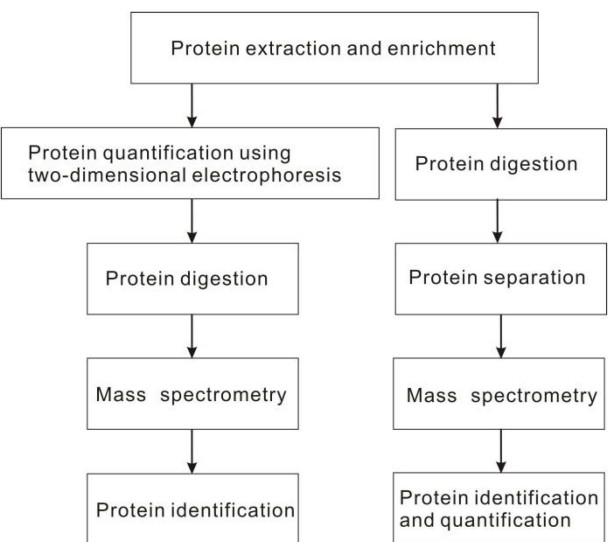

**Figure 5.** Two most common pipelines for quantitative proteome analysis.

Proteomics also identifies the candidate proteins underlying complex traits by linking protein expression to genetic maps through QTL analysis [153]. The identified proteins serve as powerful biomarkers for the precision breeding of quantitative traits in plants [154]. Currently, proteomics and associated QTL mapping have successfully identified functional proteins and genes related to production and stress tolerance in several crops [155]. For instance, the large-scale proteome sequencing of 102 barley genotypes revealed drought-sensitive proteins in the different genotypes [156]. A further genetic linkage analysis of these proteins identified several proteomic QTLs (pQTLs) with potential breeding value for drought-tolerant barley. In addition, the label-free proteome sequencing of 148 recombinant inbred lines of pepper (*Capsicum annuum*) revealed several candidate hotspot regions encoding functional proteins related to fruit development through pQTL analysis [157]. The identified proteins could serve as stable markers for further breeding processes.

However, despite its great potential in plant biology research and genetic breeding, proteomics is limited by several challenges compared to genomic and transcriptomic approaches [158]. First, the identification and quantification of the whole proteome are still challenging due to the limitations of the different proteome sequencing methods. Second, the precision and reproducibility of proteome sequencing and existing proteomics pipelines are unsatisfactory. Third, deciphering the complex proteomic networks is still challenging since proteomics is far more complex than genomics. To date, proteomics has mainly served as an ancillary strategy for functional studies in the system biology of plants. However, the large-scale applications of plant proteomics are still a long way off [159]. In the future, integrating proteomics, including single-cell and spatial proteomics, with computational approaches and other omics will enhance the proteomics potential in deepening our understanding of the functions and interactions of proteins [160,161].

### 2.5. Metabolomics

Metabolomics is an emerging post-genomics tool for comprehensive qualitative and quantitative studies of small-molecule metabolites with molar masses below 1000 in the cells or tissues [162]. Plants produce various metabolites, including primary and secondary metabolites. Primary metabolites are essential for plant growth and development, while secondary metabolites play a major role in the plant responses to environmental factors [163]. Metabolites are the end products of gene transcription and protein expression within an organism (Figure 6) and act as links between genotypes and phenotypes [164]. Among the omics tools, metabolomics has the closest relationship with the phenotype [165]. Based on this, metabolomics has become an increasingly popular system biology tool for deciphering plant science [166].

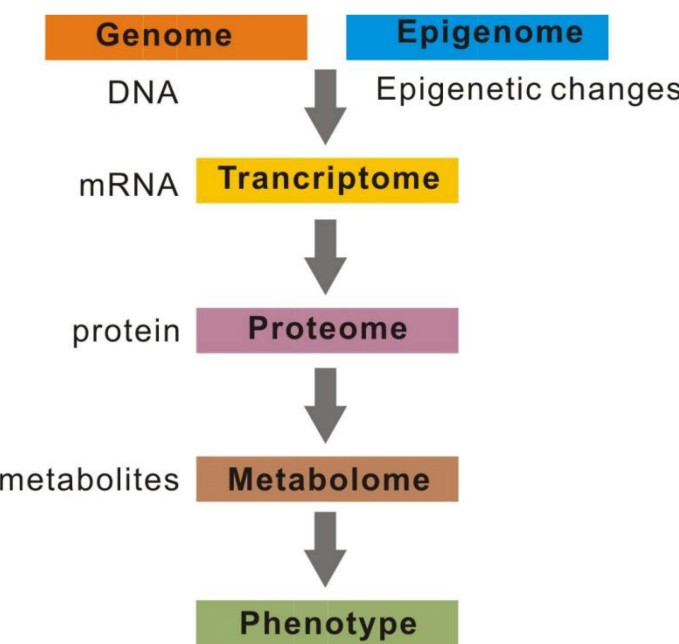

**Figure 6.** Relationships among various omics technologies.

The plant metabolome is highly dynamic and complex, with many small-molecule metabolites of diverse structures and content, making accurate metabolome profiling challenging [167]. To date, a few analytical techniques have been developed for high-throughput quantitative metabolomics, including nuclear magnetic resonance, liquid chromatography-mass spectrometry, capillary electrophoresis-mass spectrometry, and gas chromatography-mass spectrometry [168].

Metabolomics is classified into targeted and untargeted based on the study subject [169]. Targeted metabolomics performs a target analysis of known metabolites with high sensitivity and accuracy, revealing the fluctuations in specific metabolic pathways. In contrast, untargeted metabolomics performs a non-biased detection of all metabolites while identifying the differential metabolites with significant changes for further screening analysis. Similar to other omics, single-cell sequencing can be integrated into metabolomics to unravel the cellular metabolism dynamics under environmental changes [170]. Given the metabolic complexity in plants and the limitations in each analytical platform/method, combined approaches are increasingly employed in plant metabolome profiling studies.

The variation in metabolites modulates diverse biological processes and may alter plant phenotypes [171]. As a result, transcriptomics and metabolomics-based correlation analyses have enabled the genome-wide discovery of key genes controlling known and new metabolic pathways [172]. This strategy has been regularly applied in forest plants, including *Zanthoxylum armatum* [109], *Phyllostachys edulis* [173], *Populus tomentosa* [174], and *Hevea brasiliensis* [175]. Given that most metabolic traits are heritable across gener-

ations [176], combining metabolomics with QTL and GWAS can establish direct links between metabolites and phenotypes based on large-scale population metabolomic and phenotypic data [166]. To date, most metabolome QTL (mQTL) and metabolome-based GWAS (mGWAS) have been applied in hunting genes related to metabolic traits in crop plants [166], with a few success cases in poplars and apple trees. For example, a mQTL analysis of the untargeted metabolic profiling data and genetic linkage maps revealed mQTL hotspots with many peel- and flesh-related metabolites [177]. In addition, the mGWAS analysis for flavonoid features in *Popuous tomentosa* using targeted metabolomics data revealed that more than 1500 significant associations accounted for phenotypic variation [178]. Thus, the metabolic profiling of plant species can provide key functional gene information on metabolite markers linked to specific phenotypes and downstream molecular marker design [166].

Metabolic markers can be identified using metabolic profiling data under various stress conditions. The identified metabolite markers can efficiently guide further plant breeding processes for genetic improvements through direct metabolic engineering [179]. For example, the identified metabolites can help plant breeders accurately identify stress levels in plants [166]. In particular, metabolomics efficiently classified drought-tolerant eucalyptus clones into drought sensitive or tolerant based on their metabolite levels [180]. However, although metabolomics has proven to be an effective tool in plant genetic breeding programs, a single approach to identifying and quantifying all metabolites within a plant species is still lacking [168]. Therefore, future advances in metabolomics approaches should focus on more interesting gene and metabolic pathways beneficial for further plant breeding.

### 2.6. Other Omics

Molecular sequencing data and phenotypic observations are essential for the accurate interpretation of genetic architecture underlying complex traits. Plant phenomics is an emerging, interdisciplinary technique that collects high-throughput phenotypic data at an organism-wide scale using cutting-edge optical imaging techniques [181]. This image-based approach performs automated monitoring, acquisition, and processing of high-dimensional and resolution phenotypic data under various developmental and environmental conditions, with low labor consumption and error rate, unlike the traditional manual measurements. For example, unmanned aerial vehicle imagery has been used to monitor the dynamics of needle physiological traits in slash pine (*Pinus elliottii*) [182] and leaf-level, drought-related phenotypes in *Populus nigra* [183] at the population scale. The obtained standardized and highly accurate phenotypic data are useful for identifying effective genotype–phenotype relationships through QTL or GWAS analysis, which provides key information for further breeding processes.

Environmental factors play important roles in influencing plant phenotypes through genotype–environment interactions. Therefore, envirotyping for micro- or macro-environmental factors is beneficial for understanding how the complex network of environmental cues, including climate, soil, biotic, and crop management, influences plant growth and development [184]. Phenotypes are predicted more precisely by integrating spatiotemporal genotype, phenotype, and envirotype data into a training model, particularly in a changing climate [185]. Therefore, the recently developed enviromics has great potential in the smart breeding of new forest plant varieties.

In addition, plant ionomics, which studies metal, non-metal, and metalloid compositions in plants through high-throughput elemental profiling [186], can help identify genes related to plant development and stress resilience [187]. Recently, the plant microbiome was found to alter plant phenotypes by influencing their disease resistance, abiotic stress tolerance, and growth promotion [188]. Overall, the different omics tools can help us delve into the complex gene regulatory networks related to complex plant traits from different perspectives.

*2.7. Multi-Omics Integration*

Genomics is the most used omics discipline. The whole-genome DNA sequence informs the basic properties of a plant species but cannot solely determine the final phenotype. At the same time, not all DNA sequence variants lead to phenotypic variation. Instead, phenotypic plasticity is shaped by many molecular mechanisms, including epigenetic modification, gene expression and silencing, post-translational protein modification, and metabolite accumulation. Therefore, a single omics cannot sufficiently and comprehensively unravel the complex biological regulatory networks controlling the various phenotypic traits [189].

The continuous and rapid progress in developing various high-throughput omics technologies has facilitated the integration of different omics data for plant system biology studies [190]. Transcriptomics, proteomics, and metabolomics are the most frequently used omics technologies in the multi-omics integration (MOI) studies of plants, as they are the core of system biology [190]. MOI studies are accelerated by genomic information provided by well-annotated genomes and associated genomic analysis epigenomics and other omics approaches. For example, since DNA methylation alters gene expression, transcriptome analysis is combined with methylation analysis to unravel the functional relationship between the epigenome and transcriptome [122]. In particular, Balmant et al. (2020) integrated genomics, transcriptomics, and phenomics into system genetics frameworks to identify the key regulators related to lignin biosynthesis in *Populus deltoides* [191]. Generally, MOI analysis mainly involves the establishment of associations between datasets from different omics [192]. However, multiple omics platforms produce a large amount of high-throughput data, greatly challenging the subsequent MOI analysis [193]. Therefore, analysts require a good understanding of the formats and characteristics of various omics data sources and a good background in software operations, statistical modeling, and data interpretation.

The most simple and intuitive analysis strategy in MOI studies is the correlation analysis between two or more omics datasets using various models, such as Pearson, Spearman, and Kendall rank correlation analyses [194]. These correlation analyses can be applied to differentially expressed or specific biochemical pathway-related transcripts, proteins, and metabolites. However, various biological factors along with experimental errors, may cause weak correlations between different omics datasets [190,195]. For example, transcriptome and proteome sequencing of *Quercus ilex* under severe drought conditions recorded a poor correlation ($r = 0.11$) between mRNA and protein [196]. Such inconsistencies are alleviated by further sequencing and analysis.

Another strategy for analyzing MOI-related data is clustering analysis based on the similarity of various omics data using hierarchical or partition clustering methods [197]. Several statistical approaches, including similarity matrices, canonical correlation and co-inertia analysis, and matrix factorization, have been successfully applied in grouping forest plant multi-omics data [197]. For instance, Pascual et al. (2017) used a *k*-means clustering approach to integrate proteomic, metabolomic, and physiological data of *Pinus radiata* based on their quantitative trends during different periods of ultra-violet treatment, obtaining 30 clusters [198]. Further analysis can correlate the clustering results with specific scientific questions. Moreover, multivariate-based analysis has enabled the integration of multi-omics data using multi-variant data analysis approaches, such as principal components analysis, partial least squares, and orthogonal projections to latent structures (OPLS) [199]. For example, the OPLS analysis of the transcriptome, proteome, and metabolome data from transgenic poplars identified several proteins related to wood formation [200].

A common weakness of correlation, clustering, and multivariate-based analyses is that they are based on statistical methods rather than prior knowledge of molecular mechanisms. However, bioinformatics analysts have developed several pathway-based approaches, such as pathway mapping and co-expression analyses, to integrate known pathway information into MOI analysis [190]. Pathway mapping analysis maps various omics datasets against publicly available pathway databases, such as the Kyoto Encyclopedia of Genes

and Genomes (KEGG) and MetaCyc [201]. For example, López-Hidalgo et al. (2018) reconstructed and visualized 123 of the 127 known KEGG pathways in *Quercus ilex* at the transcriptome, proteome, and metabolome levels using MapMan software [202]. At the same time, multiple omics datasets could be employed to annotate certain plant metabolic pathways using known pathway information as the reference. For example, Wang et al. (2021) reconstructed the biosynthesis of two alkaloids, sanshools and wgx-50, using transcriptome and metabolome data [109].

Alternatively, the co-expression analysis can be integrated into existing pathway databases. The WGCNA approach, the most popular gene co-expression analysis tool, can detect regulatory networks for each omics layer, construct a consensus correlation network [203,204], and identify hub elements, such as hub genes, proteins, and metabolites. Furthermore, the multi-omics WGCNA approach can perform efficient gene and module clustering and provide key regulatory network information through MOI.

However, neither statistical-based nor pathway-based MOI methods are independent. Instead, these methods are often combined to answer biological questions precisely. Several complementary MOI approaches exist, such as top-down differential analysis and bottom-up genome-scale modeling [190]. These MOI approaches have been widely used in the in-depth analysis of complex metabolic pathways and other biological processes in forest plants. In particular, MOI studies provide deep insights into the genotype–phenotype–environment interactions in forest plants, essential to reconstruct complex prediction models for precision breeding [205]. Moreover, MOI studies provide a comprehensive list of functional genes and pathways for further experimental validation analysis, accelerating genome/epigenome editing for the generation of new forest plant varieties. However, the heterogeneity in signal-to-noise ratio across multiple omics layers, the adverse effects of missing values, the limitation in the interpretation ability of multi-omics models, and data sharing difficulties, such as metadata annotation, data storage, and computing resources, remain universal challenges across all-pervading MOI studies [193]. Nonetheless, MOI analysis is a powerful tool for genome-wide functional element identification and forest plant breeding. Furthermore, the development of single-cell sequencing technology provides an exciting opportunity for single-cell MOI analysis.

## 3. Conclusions and Prospects

Forest plants, including overstory trees and understory shrubs and herbs, are recalcitrant and orphan plant species. Most forest plants, except several woody plant models, are less studied than crop plants since most are inedible. However, as the population grows and climate change intensifies, forest plants provide a better ecosystem and economic services. Thus, with the improvement of people's living standards and the increasing global demand for wood, forest plant breeding for cultivars with higher wood production and greater resilience to climate change has received worldwide attention. Compared to conventional phenotype-based breeding, such as MAS and GS, plant breeding in the omics era provides a more efficient tool for genetically improving many objective traits across many plant species. Multi-omics techniques are highly innovative and still evolving, leading to a higher resolution and a deeper understanding of complex traits in forest plants. Forest plant breeding using omics technologies is greatly promoted by the rapid development of high-throughput platforms and successful omics-based breeding in crop plants. Moreover, given that most forest plant breeding programs do not result in edible products, forest plant breeders do not usually have to worry about the safety of genetically modified foods. The omics technologies, including genomics, transcriptomics, epigenomics, proteomics, and metabolomics, are useful tools for the genome-wide identification of functional elements contributing to various phenotypes. The availability of a complete or nearly complete genome sequence of model and non-model forest plants facilitates downstream analysis based on the various omics technologies. Epigenomics, transcriptomics, proteomics, and metabolomics are useful tools for understanding the gene regulatory networks of forest plants at different levels. At the same time, MOI analysis has become a popular approach

for forest plant breeding as it combines the advantages of different omics technologies, accurately identifying key functional elements by establishing associations between different omics layers. The identified functional elements can be used as powerful biomarkers for the precision breeding of forest plants, which forms an important foundation for functional verification and genetic engineering studies. However, although we are in the big data era, the current MOI analysis technique is underdeveloped. Establishing effective links between different omics layers is still challenging for forest plant breeders, who require a solid omics background, competent big data processing and mining skills, and systems biology knowledge. This review provides elementary knowledge on applying multi-omics in forest plant breeding.

**Author Contributions:** M.W. and Q.Z. designed the study. M.W. and R.L. collected the references and drafted the manuscript. Q.Z. reviewed and revised the manuscript. All authors have read and agreed to the published version of the manuscript.

**Funding:** This work was supported by Sichuan Province Central Committee Guide Local Science and Technology Development Project (grant no. 2022ZYD0095) and the start-up funds provided by Chengdu University (2081921039).

**Data Availability Statement:** Not applicable.

**Conflicts of Interest:** The authors declare no conflict of interest.

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
