# Peer review of "Multi-Omics Techniques in Genetic Studies and Breeding of Forest Plants"

_forests, doi:10.3390/f14061196_

Round 1
Reviewer 1 Report
In the manuscript, the authors review the application of multiple omics in trees. The way in which the omics were presented in the text is satisfactory and it s very clear. However, the manuscript's title does not reflect what is found in the text since the article has a more academic tone than how breeders can use multiple omics to increase their program's efficiency (I suggest changing the title). For example, in the Genomics section, the authors discuss the pan-genomes (very important to understand the diversity within a species or taxonomic group). But how have breeders been translating this information into practical decisions? How is this affecting the breeders' decisions? I believe that more results related to each omic should be presented in the manuscript ( as well as their impact on the breeding process of trees). For example, how has Genomics been used to predict new materials in species of importance (e.g., in eucalyptus, poplar, spruce, pine), and what have the authors been finding in their studies? Or how has Metabolomics been used for the breeding of crops? What type of information can the breeder leverage using this type of omics in their programs? Furthermore, presenting the advantage and bottlenecks for each omic is important.
The Multi-omics Integration needs to be improved. I understand the difficulty of doing it since there are many methods, and it isn't easy to subset them. Some "multi-omics integration" passages are available inside the individual omics discussions (e.g., Epigenomics and Transcriptomics); please carefully review them and move those passages for the Multi-omics Integration. Furthermore, the authors state that Genomics is the most used omics, but there is no explicit example of multi-omics integration involving Genomics. For example, one can integrate Genomics and Transcriptomics into system genetics frameworks to decipher biological networks and pathways for complex traits (e.g., see Balmant et al. 2020 – doi: 10.1101/gr.261438.120). Multiple omics can also be integrated into prediction models and help breeders in decision-making. As pointed out before, the key message here is to show how integrating multiple omics can help the breeder efficiently deploy superior materials, not only describe methods.
I suggest the inclusion in the manuscript of two very important omics that are missing in the review and that can positively affect forest breeding (if you do not include them, at least add some paragraphs in the manuscript describing their importance and their impact for breeding): 1) Phenomics (there is a lot of work on this topic in trees); and 2) Enviromics.
Figure 1 is too generic. Please depict it more detailedly and explain more in the text. For example, show the time needed to finish one breeding cycle using traditional breeding, GS, or MAS. I suggest that the authors pick one species (e.g., eucalyptus, pinus, spruce) and depict it in more detail (similar to Resende et al. 2017 - https://doi.org/10.1038/hdy.2017.37, or Resende et al. 2012 https://doi.org/10.1111/j.1469-8137.2011.04038.x)
Author Response
In the manuscript, the authors review the application of multiple omics in trees. The way in which the omics were presented in the text is satisfactory and it s very clear.
Response: Thanks for your positive comments. We have revised the manuscript point to point according to your suggestions.
However, the manuscript's title does not reflect what is found in the text since the article has a more academic tone than how breeders can use multiple omics to increase their program's efficiency (I suggest changing the title).
Response: We have changed the title into “Multi-omics techniques in genetic studies and breeding of forest plants”.
For example, in the Genomics section, the authors discuss the pan-genomes (very important to understand the diversity within a species or taxonomic group). But how have breeders been translating this information into practical decisions? How is this affecting the breeders' decisions? I believe that more results related to each omic should be presented in the manuscript ( as well as their impact on the breeding process of trees). For example, how has Genomics been used to predict new materials in species of importance (e.g., in eucalyptus, poplar, spruce, pine), and what have the authors been finding in their studies? Or how has Metabolomics been used for the breeding of crops? What type of information can the breeder leverage using this type of omics in their programs?
Response: Thanks for your useful suggestion. We have added several sentences describing how breeders can translate information from various omics into breeding processes into each part. We have not added too much sentences since the major task of this review is to provide forest plant breeders with an elementary knowledge of multi-omics techniques.
Furthermore, presenting the advantage and bottlenecks for each omic is important.
Response: It is difficult to describe advantages of each omics since they provide different perspectives at different levels. We have added some sentences describing their limitations.
The Multi-omics Integration needs to be improved. I understand the difficulty of doing it since there are many methods, and it isn't easy to subset them. Some "multi-omics integration" passages are available inside the individual omics discussions (e.g., Epigenomics and Transcriptomics); please carefully review them and move those passages for the Multi-omics Integration. Furthermore, the authors state that Genomics is the most used omics, but there is no explicit example of multi-omics integration involving Genomics. For example, one can integrate Genomics and Transcriptomics into system genetics frameworks to decipher biological networks and pathways for complex traits (e.g., see Balmant et al. 2020 – doi: 10.1101/gr.261438.120). Multiple omics can also be integrated into prediction models and help breeders in decision-making. As pointed out before, the key message here is to show how integrating multiple omics can help the breeder efficiently deploy superior materials, not only describe methods.
Response: Thanks for your valuable suggestion. We have moved the "multi-omics integration" passages into the MOI part, and added a case study of integrating genomics and transcriptomics. We have also added the information of reconstruction of complex prediction models for precision breeding.
I suggest the inclusion in the manuscript of two very important omics that are missing in the review and that can positively affect forest breeding (if you do not include them, at least add some paragraphs in the manuscript describing their importance and their impact for breeding): 1) Phenomics (there is a lot of work on this topic in trees); and 2) Enviromics.
Response: We have added the defination phenomics, enviromics and ionomics into a new section named “2.6. other omics”.
Figure 1 is too generic. Please depict it more detailedly and explain more in the text. For example, show the time needed to finish one breeding cycle using traditional breeding, GS, or MAS. I suggest that the authors pick one species (e.g., eucalyptus, pinus, spruce) and depict it in more detail (similar to Resende et al. 2017 - https://doi.org/10.1038/hdy.2017.37, or Resende et al. 2012 https://doi.org/10.1111/j.1469-8137.2011.04038.x)
Response: Thanks for your suggestion. We have carefully read the article “Genomic Selection for Forest Tree Improvement: Methods, Achievements and Perspectives” (https://doi.org/10.3390/f11111190). As it described, the selection of Eucalyptus species usually takes 2-3 years, which is one-third of the breeding cycle (6-9 years). The GS and MAS of Eucalyptus species only takes 5 years. We have modified Figure 1 and changed its title.

Reviewer 2 Report
Dear Authors,
First of all, It would be helpful if the article was reviewed by a native speaker. I have made some suggestions and corrections in the Manuscript, please find them in the attached document. I will also have some suggestions for the manuscript;
# In the Introduction part, more advanced technologies (eg CRSPR, transformation studies etc.) can be mentioned and current examples can be given from some studies on breeding.
# References should be arranged according to the spelling rules of the journal. For example, the names and abbreviations of journals.
Kind regards

It would be helpful if the Manuscript was reviewed by a native speaker.
Author Response
First of all, It would be helpful if the article was reviewed by a native speaker.
Response: Thanks for your suggestion. The final manuscript has been reviewed by a native speaker, and the certificate has been provided as an attached document.
I have made some suggestions and corrections in the Manuscript, please find them in the attached document.
Response: We have made corrections point to point according to your suggestions provided in the attached document.
I will also have some suggestions for the manuscript;
# In the Introduction part, more advanced technologies (eg CRISPR, transformation studies etc.) can be mentioned and current examples can be given from some studies on breeding.
Response: We have added two sentences at the beginning of the last paragraph in the Introduction part, which simply introduce the advanced CRISPR, RNA interference, and virus-induced gene silencing, and their roles in plant breeding.
# References should be arranged according to the spelling rules of the journal. For example, the names and abbreviations of journals.
Response: We have checked through all references and made corrections to ensure that all references have been arranged according to the spelling rules of MDPI journals.

Round 2
Reviewer 1 Report
The authors have answered and performed all possible suggestions I made (and also the suggestions made by the other reviewer). I liked the new information added to the manuscript, and the change in the title represents better what a reader will find in the article. The changes in Figure 1 shows, in a much clear way, the advantage of using Genomic Prediction to accelerate the development of new varieties in forest breeding programs.
The only suggestion I have is to carefully check the manuscript one more time and look for typos; for example, in the first paragraph of the Introduction, one can find "environmen-friendly". Furthermore, please use "new tree varieties" instead of "new forest plant varieties."